# An Artificial Neural Network Based on Oxide Synaptic Transistor for Accurate and Robust Image Recognition

**DOI:** 10.3390/mi15040433

**Published:** 2024-03-24

**Authors:** Dongyue Su, Xiaoci Liang, Di Geng, Qian Wu, Baiquan Liu, Chuan Liu

**Affiliations:** 1The State Key Laboratory of Optoelectronic Materials and Technologies, Guangdong Province Key Laboratory of Display Material and Technology, School of Electronics and Information Technology, Sun Yat-sen University, Guangzhou 510275, China; sudy5@mail2.sysu.edu.cn (D.S.); liubq33@mail.sysu.edu.cn (B.L.); liuchuan5@mail.sysu.edu.cn (C.L.); 2State Key Laboratory of Microelectronic Devices and Integrated Technology, Institute of Microelectronics, Chinese Academy of Sciences, Beijing 100029, China; digeng@ime.ac.cn; 3School of Computer and Information Engineering, Guangdong Polytechnic of Industry and Commerce, Guangzhou 510510, China; wuqian1427@gdgm.edu.cn

**Keywords:** synaptic transistors, artificial neural network, image recognition

## Abstract

Synaptic transistors with low-temperature, solution-processed dielectric films have demonstrated programmable conductance, and therefore potential applications in hardware artificial neural networks for recognizing noisy images. Here, we engineered AlO_x_/InO_x_ synaptic transistors via a solution process to instantiate neural networks. The transistors show long-term potentiation under appropriate gate voltage pulses. The artificial neural network, consisting of one input layer and one output layer, was constructed using 9 × 3 synaptic transistors. By programming the calculated weight, the hardware network can recognize 3 × 3 pixel images of characters z, v and n with a high accuracy of 85%, even with 40% noise. This work demonstrates that metal-oxide transistors, which exhibit significant long-term potentiation of conductance, can be used for the accurate recognition of noisy images.

## 1. Introduction

Artificial neural networks (ANNs) have achieved remarkable success in machine vision tasks such as image recognition, driving the demand for specialized hardware [1,2,3,4]. Emerging devices combining storage and computational functionalities have become the focus of research, including memristors [5,6,7], synaptic transistors [8,9,10,11], and ferroelectric transistors [12,13,14,15]. By emulating their behavior, it is possible to instantiate neural networks capable of executing complex tasks like image recognition directly at the hardware level. However, the practical implementation of neuromorphic computing using these in-memory computation devices presents significant challenges, particularly maintaining robust operation under non-ideal environmental conditions.

Oxide-based synaptic transistors are promising candidates for the implementation of ANNs [10]. They leverage the formation of the electric double layer in the oxide electrolyte that enables substantial modulation of the channel conductance, thereby mimicking the biological synaptic plasticity [16,17,18]. This property means they are suitable for use as weights in connections of the network, especially when dealing with high-noise image recognition. Furthermore, amorphous oxide films are attractive because they are compatible with large-area semiconductor device processes that can be integrated into device networks.

In this study, we employed an InO_x_ synaptic transistor with an AlO_x_ solid electrolyte as the dielectric layer and 9 × 3 transistors were used to construct a single-layer neural network model. The synaptic transistor exhibited pronounced long-term potentiation, which enables the construction of networks and recognizes images under high-noise conditions.

## 2. Materials and Methods

The method of film deposition has been reported in a previous study [19]. The AlO_x_ precursor was synthesized by dissolving aluminum nitrate hydrate, nitric acid and ammonium hydroxide in hydrogen peroxide, and it was spin-coated on a heavily doped silicon wafer. The sample was annealed on a hot plate at 300 °C for 30 min. After repeating this process five times, the thickness of the AlO_x_ film reached about 30 nm. Then, the InO_x_ precursor was synthesized by dissolving indium nitrate hydrate in deionized water and was spin-coated on the AlO_x_ film. The sample was annealed on a hot plate at 230 °C for 2 h. The Al top electrodes were deposited on the InO_x_ film to fabricate an AlO_x_/InO_x_ capacitor. For the fabrication of the transistor, the InO_x_ film was patterned by photolithography and etched with hydrochloric acid. The InO_x_ films were patterned to reduce the gate leakage current. Subsequently, the sample was annealed on a hot plate at 100 °C for 10 min to repair the damage from the etching process. Then, Al was used as source/drain electrodes to fabricate bottom-gate, top-contact transistors. The films and devices were characterized by a semiconductor parameter analyzer and an electrochemical impedance analyzer. The composition of the SiO_2_/AlO_x_ thin film was characterized by secondary-ion mass spectroscopy (SIMS), which measured the depth profiles with a 2 kV Cs+ sputter beam. Cross-sectional transmission electron microscope (TEM) images were obtained with the JEM 2100F system operating at 200 kV and the samples were prepared on silicon using focused-ion-beam techniques.

## 3. Results

Figure 1a shows the scheme of the AlO_x_/InO_x_ capacitor. The capacitor is equivalent to the circuit that contains three parallel RC in series, representing the dielectric layer, electric double layer (EDL) and semiconductor layer. The EDL is formed by the accumulation of protons at the interface between AlO_x_ and InO_x_ under positive bias at the bottom electrode. Thus, the impedance and the capacitance are dependent on the frequency and the bottom electrode bias, as shown in Figure 1b,c. The measured capacitances were determined by the complex impedance and represent the response of the film to the AC voltage with various frequencies and DC biases. The capacitive components in the equivalent circuit represent the corresponding charge storage and release processes within the thin film. For example, the EDL capacitance *C*_EDL_ represents the storage and release of ions at the interface between AlO_x_ and InO_x_. Due to the formation of the EDL, the total capacitance of the capacitor increases as the bias increases. Meanwhile, induced by the EDL, the carriers in InO_x_ accumulate at the interface and increase the conductance of InO_x_. Thus, the total impedance of the capacitor decreases as the bias increases. At negative bias, due to the absence of EDL formation, the impedance and capacitance are close at bias values of 0 V and −3 V. 

Therefore, the conductance of the transistor can be modulated in a wide range by *V*_g_ pulse stimulation, as shown in Figure 1e. The gate leakage current at *V*_g_ = 2 V is about 2.9 × 10^−9^ A, which is lower than *I*_d_ = 6.3 × 10^−5^ A. The results can confirm that the electrical insulation characteristics of the AlO_x_ layer are good enough for the gate dielectric layer. This is because the capacitance increases when the frequency decreases at a bias value of 3 V, as shown in Figure 1c. It indicates the interfacial ion concentration would increase with a reducing scan rate. Corresponding to the low frequency (0.1–10 Hz) with large capacitance, an appropriate scanning rate (in Figure 1e, the scan rate is 0.41 V/s) can effectively ensure the formation of EDL and the plasticity of the device. Under a positive gate voltage, the ions in AlO_x_ move towards the interface. Due to the ion accumulation, the electric double layer (EDL) is formed near the interface. In addition, the ions may be adsorbed on the interface electrochemically. The large capacitance of the EDL and the charge from adsorption stimulate the channel carrier and increase the current. Depending on the decay time of the accumulated ions, the increased current can be maintained for a long time (serving as long-term potentiation) or a short time (serving as short-term potentiation). By applying the *V*_g_ pulses, the current increases with the continuous pulse stimulation, demonstrating short-term potentiation, as shown in Figure 1f. After the pulses, the current decays but remains above the initial value, indicating long-term potentiation. Various pulse widths and intervals would affect the potentiation behaviors. As shown in Figure 1g, when increasing the pulse width from 20 ms to 70 ms, the drain current stimulated by 30 gate pulses increases from 0.72 μA to 1.10 μA. As shown in Figure 1h, when increasing the pulse interval from 20 ms to 70 ms, the drain current stimulated by 30 gate pulses decreases from 0.52 μA to 0.37 μA. By increasing the pulse width and reducing the pulse interval, the accumulated ions at the interface increased, resulting in a higher drain current. The paired pulse facilitation (PPF) behavior can be observed in Figure 1i. The PPF index can be fitted by the double-phase exponential function as $PPF\;index=1+A_{1}\exp\left(-∆t/\tau_{1}\right)+A_{2}\exp\left(-∆t/\tau_{2}\right)$. $\tau_{1}$ and $\tau_{2}$ are estimated to be 12.3 ms and 284.3 ms. The relaxation time indicates that the PPF is mainly related to ion response at 1–10 Hz.

To characterize the hydrogen in the alumina, transmission electron microscopy (TEM) of Si/AlO_x_/InO_x_ and secondary ion mass spectrometry (SIMS) of SiO_2_/AlO_x_ were performed. As shown in Figure 2a, the alumina was amorphous with inapparent pinholes. The pinholes are probably formed due to the decomposition of nitric acid and ammonium hydroxide in the precursor. The amorphous structure may provide a transport channel for the hydrogen in the film. The hydrogen concentration was related to the annealing temperature. Figure 2b,c show the hydrogen and aluminum intensity with the film annealed at 200 °C and 300 °C. The hydrogen concentration decreases when the annealing temperature increases from 200 °C to 300 °C. It indicates that the source of hydrogen ions is probably the residual decomposition of the precursor. The absorbed moisture is also a possible source because at the start of sputtering, the surface shows an obvious hydrogen intensity. This means that the moisture in the atmosphere was adsorbed by the film. To characterize the influence of temperature on the retention time, we analyzed the drain current stimulated by a gate pulse at various temperatures as shown in Figure 2d. The decay of the current was fitted with the exponential function, i.e., $I_{d}=I_{d1}\exp\left(-t/\tau \right)+I_{d0}$. The extracted time constant τ as a function of the temperature is shown in Figure 2e. When increasing the temperature from −50 °C to 50 °C, τ decreases from 2.41 s to 0.69 s. As the temperature increases, the ions migrate more easily. Thus, the carrier induced by the electric double layer decays faster after the pulse at a high temperature, resulting in a shorter retention time for the conductance.

The ability of long-term potentiation to provide the weight update function is essential in artificial neural networks (ANNs). We built an ANN that can recognize three types of images with 3 × 3 pixels like characters z, v and n. The training and test datasets contain 9999 and 999 sets of pixel data, respectively. The framework of the network contains one input layer (nine inputs) and one output layer (three outputs), as shown in Figure 3a. The input layer and output layer are fully connected; this means that the output value $o_{n}$ is the sum of all inputs, i.e., $o_{n}=\sum_{m=1}^{9}\left(p_{m}\times G_{m,n}\right)$, where $p_{m}$ is the normalized grayscale values used as input and $G_{m,n}$ is the connection weight. Considering that the conductance of a transistor is positive, $G_{m,n}$ is restricted to being greater than zero. The activation function is $y_{n}=1/\left[1+exp\left(-o_{n}\right)\right]$ (Sigmoid function). The pixel grayscale was introduced with noise, which is expressed as $p_{m}=1-\sigma p_{n}\;and\;0+\sigma p_{n}$ for the black and white pixel, respectively, where σ is the degree of noise and $p_{n}$ is the random noise in the range of zero to one. Through training the ANN in software using the backpropagation algorithm [20], the weights in the ANN are updated from the random values (Figure 3b) to converge (Figure 3c). The accuracy is 100% for the images when σ=0.4. The weight mapping indicates that three transistors are needed to update the weight.

For implementation in devices, the normalized grayscale values $p_{m}$ were converted to the drain voltage, when $0{<p}_{m}<0.1$, $V_{d}=0.1\;V$, and when $0.9{<p}_{m}<1$, $V_{d}=1\;V$. The conductance of the transistors was used as the connection weight. The sum of the drain currents can represent the output value, i.e., $o_{n}=\sum_{m=1}^{9}\left(V_{d}\times G_{m,n}\right)=\sum_{m=1}^{9}I_{dm,n}$, as shown in Figure 4a. By comparing the values of *o*_1_, *o*_2_, and *o*_3_, the images can be recognized. According to the simulation results, the corresponding devices can be trained using gate voltage pulses. The initial conductance mapping was carried out in the range of about 10 nS to 80 nS (Figure 4b) and updated to a similar conductance distribution. The corresponding conductance was in the range of about 1 μS to 5 μS (Figure 4c). Although the device size is relatively large, it can be further scaled down after optimizing the following process: (1) the spin-coating process of the InO_x_ film for improving the film’s quality and uniformity; (2) the photolithography process for high exposure resolution; (3) the etching process to eliminate the nonideal etching profile, like undercut and overetching; (4) the metal lift-off process to achieve a high yield rate.

A detailed training process for hardware device arrays was implemented in a similar way to the simulation in the software. The drain current with a period time of 3 s served as one epoch. The devices were set to an initial state of about 10 nS. When the image pixel data were input, namely the corresponding drain bias, the *I*_d_-*t* curves were measured. For the transistors that did not require updated conductance, no gate pulse was applied. For the transistors that required updated conductance, gate pulses (width: 90 ms, amplitude: 3 V) were applied. Examples of the *I*_d_-*t* curves at various drain voltages are shown in Figure 5a. Figure 5b,c show the process of recognizing image z. The grayscale values were inputted as *V*_d_ into each column, and the *I*_d_ of each row were summed to obtain *o*_n._ After 30 epochs, the conductance of the corresponding devices was enhanced to satisfy the need for image recognition, as shown in Figure 5d. 

Figure 6a shows the recognition accuracy when the images have various noises. When increasing the conductance of corresponding devices, the accuracy increases with each epoch. The accuracy can reach 100% by epoch 4 for these three images without noise (Figure 6a, black curve). With an increasing noise degree σ, the accuracy decreases. When σ=0.4, the accuracy can reach 85% by epoch 19 (Figure 6a, purple curve). The decrease in the accuracy with σ is probably because of the variation in conductance. The effect of the Δ*G* on accuracy has been estimated as shown in Figure 6b. The definition of Δ*G* in Figure 6b is the ratio of the conductance with inputting gate voltage pulses and the conductance without inputting gate voltage pulses. The accuracy tends to increase with Δ*G* when it is below 100, and then it remains almost constant. Figure 6c–e depict the evolution of the output, namely the summed current, over the course of training. When the input image is z, the corresponding summed current *o*_1_ increases with training and surpasses the *o*_2_ and *o*_3_, indicating successful recognition of image z. With the same weight mapping, the corresponding summed current can increase to the maximum among the three outputs when either image v or n is inputted.

In addition to the implementation of image recognition, conductance can be used to map characters. The conductance at appropriate locations was modulated by gate pulses (amplitude: 10 V, width: 50 ms) and read at *V*_d_ = 0.1 V after 10 s. The 3 × 3 devices can represent images such as z, v, and n (Figure 7). The contrast ratio, namely the current ratio, can attain a value of about 10^4^.

## 4. Conclusions

The synaptic transistors with AlO_x_ as the electrolyte layer and InO_x_ as the semiconductor layer were used to implement an ANN for image recognition. Through controlling the ion accumulation at the interface by gate voltage, the conductance of the AlO_x_/InO_x_ transistors shows long-term potentiation. We implemented an ANN comprising 9 × 3 synaptic transistors. By constructing appropriate weight maps, the network shows a high accuracy of 85% in recognizing three types of 3 × 3 pixel images with a high noise of 40%. This work presents a feasible approach for oxide synaptic transistors to construct a network and recognize noisy images.

## Figures and Tables

**Figure 1 micromachines-15-00433-f001:**
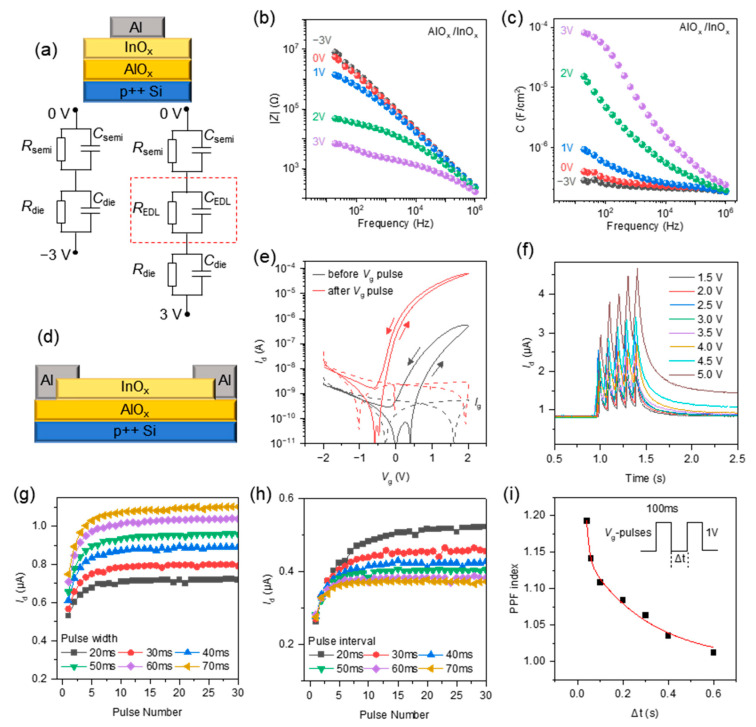
(**a**) A scheme of the AlO_x_/InO_x_ capacitor. (**b**,**c**) The impedance and the capacitance of the AlO_x_/InO_x_ capacitor at various biases. (**d**) A scheme of the AlO_x_/InO_x_ TFT. (**e**) Transfer curves of the AlO_x_/InO_x_ TFT before and after *V*_g_ pulses at *V*_d_ = 2 V. (**f**) The response of five pulses (amplitude: 1.5 V to 5 V, width: 20 ms, period: 100 ms). The drain current corresponding to the consecutive pulses with (**g**) various pulse widths (20 ms to 70 ms) and constant pulse intervals and amplitudes, or with (**h**) various pulse intervals (20 ms to 70 ms) and constant pulse widths and amplitudes. (**i**) Paired pulse facilitation (PPF) index as a function of the interval time Δt.

**Figure 2 micromachines-15-00433-f002:**
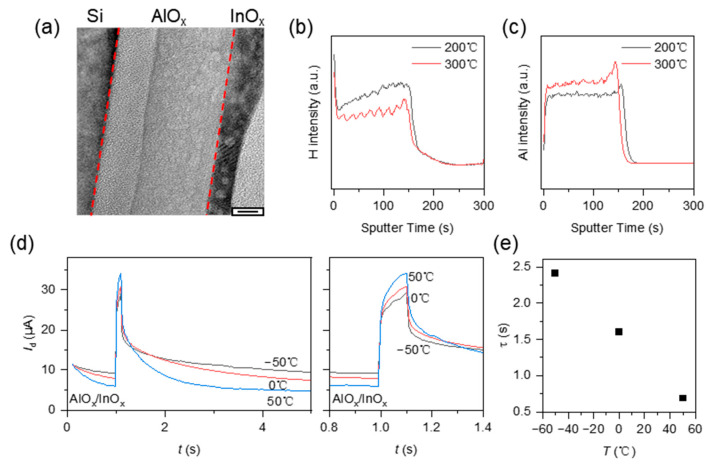
(**a**) The transmission electron microscopy (TEM) cross-section image of Si/AlO_x_/InO_x_ (scale bar is 6 nm). The secondary ion mass spectrometry (SIMS) depth profile of SiO_2_/AlO_x_ annealed at 200 °C and 300 °C for hydrogen (**b**) and aluminum (**c**). (**d**) The *I*_d_-*t* curve stimulated by a gate pulse (6 V, 100 ms) at various temperatures (left) and the enlarged view of the pulse response (right). (**e**) The time constant τ as a function of the temperature. τ is fitted by the equation $I_{d}=I_{d1}\exp\left(-t/\tau \right)+I_{d0}$.

**Figure 3 micromachines-15-00433-f003:**
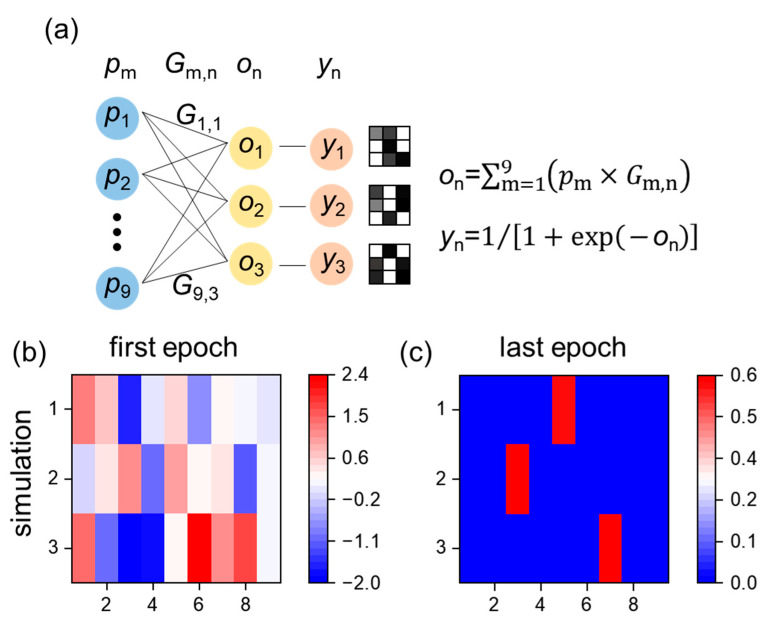
(**a**) A scheme of an artificial neural network with one input layer and one output layer to recognize the 3 × 3 pixel images z, v and n. The distribution of weights after software training at the first epoch (**b**) and the last epoch (**c**).

**Figure 4 micromachines-15-00433-f004:**
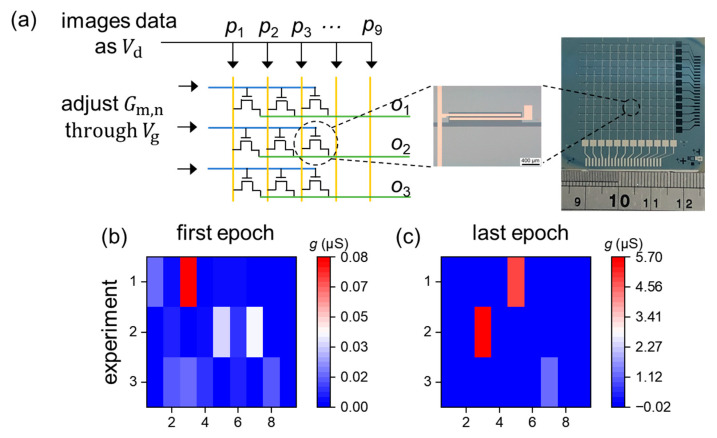
(**a**) A scheme of the 9 × 3 transistor network. The normalized grayscale values of image pixels are used as the *V*_d_. The conductance of the transistors represents the weights in neural connection and is modifiable by the gate voltage pulses. The inset shows the optical microscope photograph of the transistor array (the unit of the ruler is cm) and a transistor in the array (the scale bar is 400 μm). The distribution of weights in the transistors is illustrated for the first epoch (**b**) and the last epoch (**c**).

**Figure 5 micromachines-15-00433-f005:**
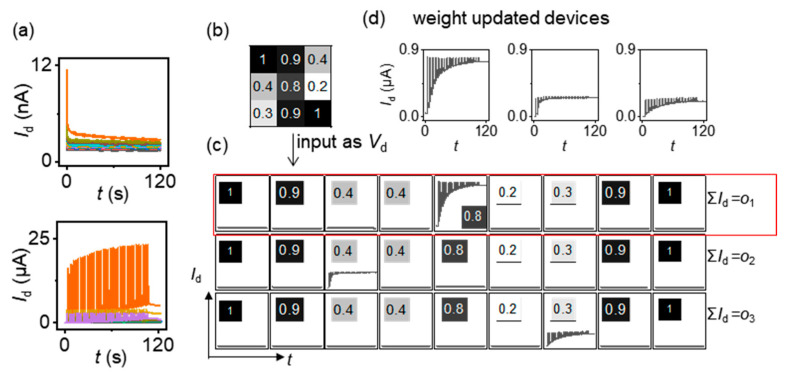
(**a**) The *I*_d_-*t* curves at various *V*_d_ with and without gate pulse stimulation. (**b**) The image grayscale matrix. (**c**) *I*_d_-*t* curves of the 9 × 3 device array. The *V*_d_ values correspond to the grayscale in (**b**). (**d**) The zoom in graphs of the weight updated devices in (**c**).

**Figure 6 micromachines-15-00433-f006:**
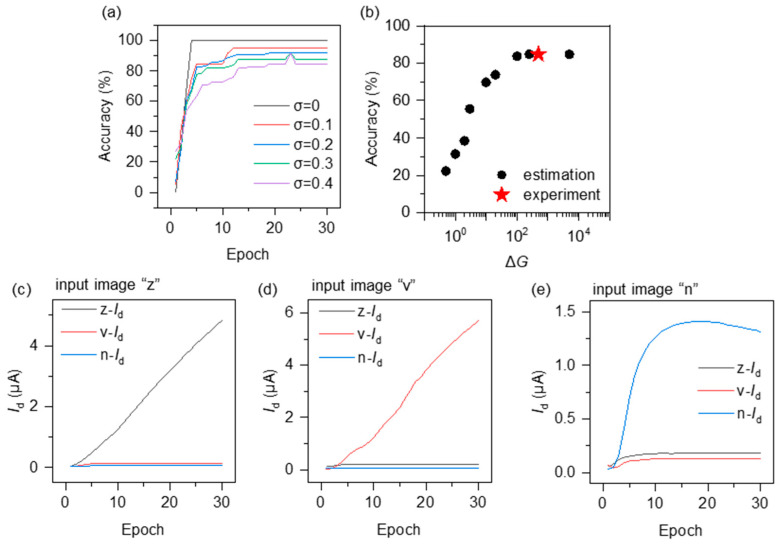
(**a**) The accuracy of image recognition at various noises *σ*. (**b**) The estimated effect of Δ*G* on accuracy at *σ* = 0.4. (**c**–**e**) Examples of input images z, v, and n, where the current z-*I*_d_, v-*I*_d_, and n-*I*_d_ represent the outputs *o*_1_, *o*_2_, and *o*_3_ in Figure 3a, respectively. The largest current indicates that the recognition result corresponds to the respective image.

**Figure 7 micromachines-15-00433-f007:**
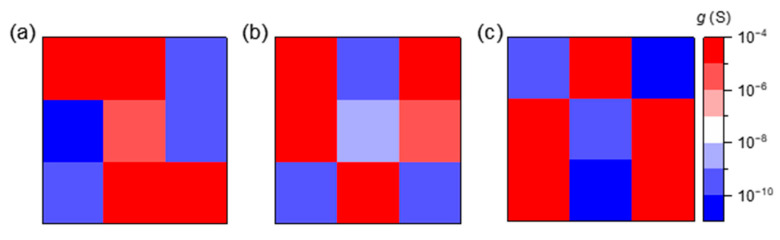
The programming conductance mapping of 3 × 3 AlO_x_/InO_x_ TFTs representing the characters (**a**) z, (**b**) v and (**c**) n. The scale bar is ranged from 10^−10^ S (blue) to 10^−4^ S (red). The red pixels represent devices with high conductance, which form the simple characters in 3 × 3 pixel images.

## Data Availability

The data presented in this study are available from the corresponding author upon reasonable request.

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
