# Peer review of "An Artificial Neural Network Based on Oxide Synaptic Transistor for Accurate and Robust Image Recognition"

_micromachines, 2024, doi:10.3390/mi15040433_

Round 1

Reviewer 1 Report

Comments and Suggestions for Authors

In this study, the authors have fabricated AlOx/InOx synaptic transistors using a solution processing method to constitute neural networks with the capability of image recognition. The researchers achieved to program the network to recognize 3x3-pixel representations of the characters 'z', 'v', and 'n', attaining an accuracy of 85% even under substantial noise interference of 40%. This investigation enables the potential of metal-oxide transistors in precise image processing and recognition amidst notable noise challenges. This is an interesting work and I recommend it to be published after minor correction. However, a few areas require additional clarification:

1.      Could the authors discuss the connection between the capacitance represented in the equivalent circuit and the capacitance measurements displayed in Figure 1?

2.      It is noted that both impedance and capacitance in Figure 1 depend on frequency. The authors should expound on how this frequency dependence influences the plasticity of the device and provide the scanning rate employed for the transfer characteristic curve.

3.      In Figure 5b, the term ΔG requires definition. The authors are requested to clarify its meaning.

4.      The manuscript should specify the pulse conditions that were utilized to modulate the conductance as shown in Figure 6.

Reviewer 2 Report

Comments and Suggestions for Authors

Su et al. demonstrated the utilization of oxide-based synaptic transistors for constructing an artificial neural network (ANN) aimed at accurate image recognition, particularly in noisy environments. The transistors, based on AlOx/InOx, exhibit long-term potentiation, allowing for effective network construction. The ANN, consisting of 9x3 synaptic transistors, demonstrated recognition of 3x3-pixel images with a high accuracy of 85%, even in the presence of 40% noise. This work highlights the potential of metal-oxide transistors for robust image recognition despite noisy conditions. The methodology involved deposition and fabrication processes, along with training algorithms for weight adjustments. However, there are some major problems to be addressed by authors, and I can’t recommend this manuscript to be published in the journal of Micromachines with current version. The comments are following:

1.     The authors claimed that AlOx/InOx oxide transistor can be utilized for artificial synaptic devices and showed the pattern recognition results. However, they didn’t show up the enough electrical and synaptic behaviors, such as endurance, retention, LTP/LTD curves, post-synaptic behaviors depending on different pulse widths and interval timing, and retention characteristics under different pulse schemes. Authors must measure those characteristics.

2.     The authors showed the switching characteristics before and after the pulses. What mechanisms are involved with this phenomenon?

3.     The authors must provide the gate leakage current to confirm the AlOx layer is good enough for gate dielectric layer.

4.     In device fabrication process, authors mentioned that the InOx film was patterned by photolithography and etching with hydrochloric acid, and then Al was deposited as source/drain electrodes. Why should the etching be needed? And, if the electrode metals were changed to noble metal, like Au, Pd, or Pt, is the switching phenomenon still observed?

5.     Figure 2 caption is same with Figure 3 caption. The authors must revise them.

6.     Authors should provide the optical images of 3x3 oxide transistor array to give the authors comprehensive understanding of device.

Reviewer 3 Report

Comments and Suggestions for Authors

Application  of the electronic conductivity in InOx at the interface  with  AlOx due to Debye layer formation is suggested for building  synaptic elements for  neuromorphic designs.  The idea and implementation are novel and can be published.

Comments:

1.Please, add  information on the nature of protons  in alumina. Is alumina porous?  Are the ions connected with absorbed moisture?  To what extent temperature would influence the retention times ?

2. The transistors are large (~2mm length). What about scaling down? What are the limiters ? For example, can the channel length be scaled down if dry etch would be used...? Please, discuss possibility of scaling down in the discussion section.

Round 2

Reviewer 2 Report

Comments and Suggestions for Authors

The authors addressed all of my concerns and I think it would be ready to be published.